# The Relationship Between the Perceived Personality Traits of Dentists, Dental Anxiety, Negative Stories, and Negative Experiences with Dental Treatment: A Cross-Sectional Study

**DOI:** 10.3390/dj13040162

**Published:** 2025-04-10

**Authors:** Martine Eide Furnes, Sondre Lillejord, Vetle Lillejord, Jan-Are K. Johnsen

**Affiliations:** Department of Clinical Dentistry, Faculty of Health Sciences, UiT The Arctic University of Norway, 9037 Tromsø, Norway

**Keywords:** dental anxiety, personality, social perception, dentist–patient relations, public health

## Abstract

**Background/Objectives:** Negative stories about dentists are prevalent in mass media and in social discourse. This study explores the associations between dental anxiety, negative dental stories, and negative dental experiences, and how these phenomena are related to the perception of dentists’ personality traits. **Methods:** An anonymous electronic survey was distributed among students at UiT The Arctic University of Norway, and 118 students chose to participate in the study. The questionnaire included questions on dental anxiety, negative stories about dental treatment, negative experiences of dental treatment, and ratings of dentists’ personality traits. **Results:** Almost half of the participants (46%) had at least one negative dental experience, while the majority (80%) of the participants had heard a negative story about the dentist. There were significant differences in the ratings of dentists’ personality traits based on having previous negative dental experiences and based on dental anxiety scores. Participants without negative dental experiences and with low dental anxiety rated the dentist more favorably in terms of personality traits than anxious participants or participants with negative dental experiences. Negative dental treatment experiences (OR 10.35; *p* < 0.001) and rating dentists as having low extraversion (OR 0.34; *p* < 0.05) emerged as the most important predictors of high dental anxiety. **Conclusions:** The interplay between negative stories about the dentist, dental anxiety, and negative dental experiences should be explored further, as well as the impact of the perceived personality traits of dental health professionals.

## 1. Introduction

Despite evidence that dentists, by large, are trusted by the general public [1,2], negative stories about dentists and dentistry are prevalent in mass media and in social discourse. Studies have identified that dentists occupy the role of morally challenged characters in a host of different media, including movies and literature [3,4], and that the nature of dental diseases appear to be misrepresented in the case of reality tv [5]. Consequently, questions have been asked about the need to boost dentistry’s public image and about the impact of such phenomena [6].

Although seemingly trivial in nature, prevalent negative stories and stereotypes about dentists and dentistry can influence different aspects of how patients interact with the dental healthcare services. While such aspects could serve to erode trust in the dental health profession on a general level, it is easy to imagine that they could have a specific impact on issues such as dental anxiety and the avoidance of necessary dental treatment. For instance, vicarious learning from the stories shared by others or by the media has been identified as an exogenous factor by which dental anxiety is learnt or reinforced [7]. Since dental anxiety is reliably associated with negative dental treatment experiences [8,9], interestingly, this could indicate a negative feedback loop in which negative stories are reaffirmed and strengthened in light of actual negative experiences, which again paves the way for more negative future experiences. It is well documented that negative information and bad events are highly informative to human motivation and are resistant to change even in light of disconfirming evidence [10], and this also holds for negative stereotypes which are seemingly automatically reinforced without external input [11]. Further, since negative stories about dentists often point to flaws of character or morality as being central to their failures as health professionals [3], such stories should have predictable impacts on how dentists are perceived and evaluated with regard to their personal characteristics or personality traits. While dentists appear to possess the necessary personality styles to form health-promoting dentist–patient relationships [12], and dentists and patients generally appear to agree upon what aspects of personality characterize the dental profession [13,14], it remains unclear how patients’ experiences of negative stories specifically interact with their perception of the personality of dentists. It is known, however, that dentists themselves believe that they are evaluated to a large extent based on their personality [15], and that patients value interpersonal or emotional skills when choosing a dentist [16], which is supported by findings from studies on patient satisfaction in dentistry [17,18].

In light of this, the current study investigates how the personality ratings of dentists are associated with participants’ dental anxiety, personal negative experiences during dental treatment, and negative stories about dentists. The study hypothesis is that personality ratings are negatively associated with dental anxiety, negative experiences, and negatives stories, and specifically that (A) participants who have had negative dental experiences will rate dentists more unfavorably in terms of personality traits than participants without negative dental experiences; (B) participants who score highly on dental anxiety will rate dentists more unfavorably in terms of personality traits than participants with low dental anxiety; and (C) participants who have been exposed to negative stories about the dentist or dental treatment are more likely to rate dentists more unfavorably in terms of personality traits than participants who have not heard negative stories about the dentist. Also, in this study, we wanted to explore the relative importance of negative experiences, negative stories, and inferred personality traits with regard to dental anxiety, as we have argued here for the existence of a negative feedback loop related to negative stories, negative treatment experiences, and evaluations of personality on dental anxiety.

## 2. Materials and Methods

An electronic questionnaire was distributed to students at UiT The Arctic University of Norway by UiT administrative personnel. The questionnaire was in the Norwegian language and was hosted by Questback. It contained a total of 13 questions and took about 5 min to complete. The questionnaire asked for participants’ age and gender (female/male). Age was indicated on a four-point scale consisting of the following age groups: 18–25, 26–34, 35–40, and 40+.

Experiences with Treatment and Exposure to Negative Stories about Dentists: The questionnaire asked participants to indicate, by yes or no answers, if they had had a negative dental experience, and if they had heard any negative stories about the dentist.

Measurement of Dental Anxiety: A Norwegian translation [19] of the Index of Dental Anxiety and Fear (IDAF-4C) [20] was used in this study to measure the level of dental anxiety among the participants. The IDAF-4C provides participants with 8 statements about dental treatment, and they have to indicate to what extent they agree or disagree with the statement on a scale from 1—disagree to 5—strongly agree. For the statistical analysis, sum scores were calculated (8–40), and the mean IDAF-4C score was used to generate a variable of “high” and “low” dental anxiety. The internal reliability of the IDAF-4C measure was satisfactory, with Cronbach’s alpha = 0.91.

Dispositional Attributions of Dentists’ Personality Traits: To enable participants to rate the personality of dentists, a modified version of the Ten Item Personality Inventory (TIPI) was used [21]. The TIPI is a simplified measure of the big five theory of personality, measuring the following five personality dimensions: Extraversion (one’s level of sociability and enthusiasm), Agreeableness (one’s level of friendliness and kindness), Conscientiousness (one’s level of organization and work ethic), Emotional Stability (one’s level of calmness and tranquility), and Openness to Experience (one’s level of creativity and curiosity). These dimensions are represented in the TIPI as two antonyms signifying the existence of opposing dimensional traits. For instance, in the case of Extraversion there has to be an opposing tendency towards Introversion, and logic dictates that these opposing traits cannot coexist.

For the purpose of this study, the instructions of the TIPI were modified from being a self-rating tool to being a tool for rating another person. Similar procedures for capturing so-called “dispositional attributions” of personality have been used elsewhere [22], although the current study used a generalized approach rather than specific, descriptive scenarios. In the current approach, the participants were instructed to rate the personality traits of what they perceived as a “typical dentist” from 1 (strongly disagree) to 7 (strongly agree) using a total of 10 statements regarding the dentist’s personality. Since the TIPI consists of 10 antonym pairs yielding 5 personality factors, the calculation of the scores is performed according to strict scoring instructions [21]. For example, determining a respondent’s perception of the dentist’s emotional stability is calculated by subtracting the score on the statement “I experience the dentist as anxious and easily upset” from the score given to the opposite statement “I experience the dentists as calm and emotionally stable”, and then dividing the sum by 2. This approach was undertaken for all 5 personality traits. The antonym pairs used in this study were identical to those from the Norwegian TIPI translation [23].

Statistical Analyses: Due to lacking prior data from using a modified TIPI to measure dispositional attributions, sample size calculations were made with G*Power version 3.1.9.7 [24], with an aim to identify significant associations between the variables using χ2 tests. Expecting small-to-medium effect sizes of 0.35 and a standardized 80% power and *p*-level of 0.05, this would require 60 participants. A test of normality (Shapiro–Wilk) showed that the data concerning personality characteristics deviated from normality. Consequently, non-parametric, one-sided Mann–Whitney U-tests were used to test the specific hypotheses, while a logistic regression analysis was performed to investigate to if negative experiences, negative stories, and personality ratings of dentists could predict dental anxiety. Effect sizes were reported as rank-biserial correlations where applicable. Data were analyzed by using JASP 0.19.3.0.

This study was approved by REK (Regional Committee for Medical and Health Research Ethics; ref.no 2017/832) and NSD (Norwegian Centre for Research Data; NR 54216).

## 3. Results

One-hundred and eighteen (118) students at UiT The Arctic University of Norway answered the questionnaire. Of these, 81 were women (68.6%) and 37 were men (31.4%). The majority of the participants (78%) were in the age group of 18–25. A total of 54 of the participants (46%) answered that they had had a negative experience at the dentist, while 64 (54%) answered that they had not had a negative experience. A total of 94 participants (80%) had heard a negative story about the dentist, while 24 (20%) had never heard a negative story (Table 1).

### 3.1. Dental Anxiety, Negative Experiences, and Negative Stories

The relationships between dental anxiety, negative experiences, and negative stories are shown in Table 2. A significantly larger proportion of participants with negative experiences report high dental anxiety than participants without negative experiences; χ2 (1) = 30.24, *p* < 0.001. Also, the proportion of participants reporting high dental anxiety is significant among those reporting to have heard negative stories; χ2 (1) = 7.45, *p* < 0.01. Finally, the proportion of participants who have not heard negative stories is significantly larger among those that do not report a negative dental experience; χ2 (1) = 5.23, *p* < 0.05.

### 3.2. Dentists’ Personality Traits

The test of the research hypotheses can be seen in Table 3. Mann–Whitney U-tests showed statistically significant differences in the ratings of the personality characteristics of dentists between those that reported negative dental experiences and those that did not. Consistently, participants rated dentists less favorably in terms of personality characteristics when they reported having had negative experiences, which was in support of Hypothesis A. Also, there was a significant difference in the ratings of personality characteristics based on the level of dental anxiety (low versus high), which showed that those who reported low dental anxiety also consistently rated dentists’ personalities more favorably, in support of Hypothesis B. With regard to Hypothesis C, this was not supported, with only the Agreeableness dimension being significantly different between participants that reported to have heard negative stories and those that had not. All significant findings showed medium-to-small effect sizes.

### 3.3. Negative Experiences, Negative Stories, Dentists’ Personality Traits, and Dental Anxiety

The relative impact of negative experiences, negative stories, and ratings of dentists’ personalities as predictors of ratings of dental anxiety was explored by a logistic regression analysis with the dichotomous dental anxiety variable (low vs. high) as a dependent variable. Negative experiences (yes/no) and negative stories (yes/no) were included as categorical coefficients in the analysis, as well as gender (female vs. male). In addition, dichotomized personality ratings (high/low) were entered as coefficients in a separate step, and the analysis was run as a backwards conditional analysis since all dependent variables were associated with the outcome variable, as shown by the initial analyses.

The final model explained approximately 43% of the variance in measured dental anxiety. It showed that negative experiences emerged as the most important predictor of dental anxiety, although negative stories were also included as a predictor in the final model. Finally, the personality trait Extraversion emerged as a significant predictor in the final model, showing that rating dentists highly on this trait was linked to a decreased likelihood of experiencing high dental anxiety (Table 4).

## 4. Discussion

This study’s findings support the notion that negative experiences with dentists are consistently related to more negative ratings of dentists’ personalities. This type of dispositional attribution of stable personal traits from observations or recollections of behavior is regarded as a foundation of social interaction [25] and given the significance of negative information and experiences over positive [10], these findings are unsurprising. Furthermore, a substantial percentage of the participants (46%) reported having had a negative experience at the dentist. Also, similar consistent, negative ratings of personality traits were found with regard to high dental anxiety, which was expected given the intimate relationship between negative dental treatment experiences and the development of dental anxiety [8,9,26].

Overall, the social desirability of certain personality traits over others has been studied extensively [27,28,29], although this phenomenon would be more related to self-assessment approaches than to the methods utilized here. In the current study, the dimensions Agreeableness, Conscientiousness, Emotional Stability, and Openness to Experiences have distinct negative or less socially desirable opposites in the form of Quarrelsomeness, Carelessness, Neuroticism, and Conventionalism. All these concepts are interpretable as being less desirable in a dispositional sense (i.e., as a stable trait) than their counterparts. For instance, for a concept such as Neuroticism, the negative connotations are strong, with the concept being linked to a range of negative outcomes, including numerous health issues [30]. In the case of the Extraversion–Introversion dyad however, the theoretical assumptions are less clear. Even so, the current results show that dentists’ outgoing personalities, or behaviors interpreted as signaling such, are less likely to be associated with dental anxiety and negative dental experiences than introverted personalities or corresponding behavior. This might point to a contextual element, wherein introverted behaviors that are not negative per se are regarded as negative (or less positive) in the dental setting when enacted by a dental health professional. For instance, under the contextual constraints of the dental treatment setting, typical and clear “extraverted” behaviors might be more distinct to the patient as expressing a desire to communicate and interact, which are aspects of patient encounters which are consistently rated as being important to patient satisfaction among dental patients [16,17] and in other healthcare settings [31,32]. Interestingly, the advantages of extraversion in the workplace, for example, appear to hinge upon its connection to enthusiasm and positive emotionality [33], which are highly expressive concepts. It is worth noting that studies have found that extraversion tends to be related to higher ratings of happiness [34,35], and that acting extraverted when being introverted appears to bolster a positive affect [36,37]. Such positive findings related to extraversion strengthen the argument that extraverted behaviors and traits have high social desirability. Finally, several studies have found that patients with low extraversion or high introversion are particularly vulnerable to dental anxiety [38,39], although the relevance to the results found in the current study is unclear.

While dissimilar in methodological approach, studies from other healthcare settings have demonstrated that actual (self-rated) personality traits of healthcare professionals are meaningfully associated with patient satisfaction, mostly in longer term patient–provider relationships [40]. Interestingly, extraversion has been identified as a positive trait related to nurses’ performance [41], although it has been argued elsewhere that social competence is key to ensuring that extraversion leads to positive outcomes (e.g., adaptive performance) [42]. This role of interpersonal or social competence as a moderator of the effects of extraversion would probably also be valid in the dental setting and would be interesting to include as a variable in future studies.

In contrast, we found little evidence associating the personality ratings with hearing negative stories about the dentist (only the Agreeableness dimension showed an association with negative stories), although hearing negative stories about dentists was reported by the majority of the participants (80%). Nevertheless, it is known that negative stories about the dentist portrayed by friends, family, or mass media play a part in the development of dental fear [7] and that such stories may contribute to creating negative stereotypes of the dentist [3,43]. A challenge in the interpretation of the current results is that the contents of both negative stories and negative experiences were not detailed any further. While it is tempting to think that “negative” in this context points to a limited number of experiences or storylines, there is probably a considerable variability in how people have defined these matters. Also, the character flaws or morality issues that are often inferred in negative stories about dentists [3] may not transfer well to the “big five” personality domains utilized here, although perhaps they transfer best to the Agreeableness factor, and to its detriment, as indicated by the results.

The results of this study indicate that both negative experiences at the dentist and hearing negative stories about the dentist are associated with higher levels of dental anxiety among the participants. However, the importance of personal experiences in relation to dental anxiety is highlighted by the finding that having negative dental treatment experiences contributes more to predicting dental anxiety than having been exposed to negative stories. This is not surprising, given that direct experience is most commonly seen as a more efficient way of learning than vicarious experiences [44], of which negative stories could be considered an example. For dental health services, this can be viewed as an opportunity, since the individual dental health professional can strive to ensure positive patient experiences, even when patients fall into the category of problem-oriented visiting which may make them vulnerable to the vicious cycle of dental fear maintenance [45,46]. In the meantime, it may take a while to get rid of persistent, widespread, negative stereotypical narratives about dentists, as such phenomena appear self-perpetuating [11]. Other practical implications of these findings could perhaps be to look at ways to make dental professionals more aware of patients’ opinions and evaluations of them. While evaluations of dental health services are often based around the broad concept of patient satisfaction, one could imagine including more detailed interpersonal information such as personality evaluations into these evaluations. This type of patient feedback could, although being subjective and prone to biases, be used to promote self-reflection and professional development among dental health professionals, perhaps through the lens of reflective practice [47].

The current study was a cross-sectional study utilizing an anonymous, online questionnaire distributed to several e-mail lists at a Norwegian university and therefore uses a convenience sample susceptible to selection bias. However, the gender and age distribution of the study sample is representative with regard to what is known about the age and gender of Norwegian students of health sciences [48]. The findings do not point to causal or directional relationships, and the results should be interpreted accordingly. While there are other examples of using personality inventories to rate the perceived personality of others for research purposes [22], the implications, or accuracy, of these ratings are unknown. In the context of the current study, these ratings would most likely be a mix of generalized dispositional attributions (i.e., a composite of many experiences with dentists) or specific dispositional attributions (i.e., one’s experience of a specific dentist or specific dental encounter), although we would argue that the survey instructions are formulated so as to skew the distribution towards the former.

Human beings are constantly seeking to understand each other and to identify the dispositional traits that might best predict future encounters with others. In light of this, it is no surprise that people attribute personality traits to the health professionals that they encounter, including dentists or dental health professionals. The current results point to the interplay between such attributions and the experiences of, and attitudes related to, the clinical encounter itself. These findings could be of help to dental health professionals in order to understand patients’ opinions of them, and ultimately, in order to understand themselves better in their professional role.

## Figures and Tables

**Table 1 dentistry-13-00162-t001:** Summary of results concerning negative stories, negative experiences, and dental anxiety.

Variables	N	Yes (%)	No (%)
Negative Experiences	118	54 (45.8)	64 (54.2)
Negative Stories	118	94 (79.7)	24 (20.3)
	**Sex**	**N**	**M (SD)**
Dental Anxiety	Female	81	15.40 (7.77)
	Male	37	13.41 (7.51)
	Total	118	14.77 (7.73)

**Table 2 dentistry-13-00162-t002:** Proportions of dental anxiety, negative stories, and negative experiences..

	Dental Anxiety (DA)			Sig. ^1^
Negative Experiences	DA Low	DA High	Total	
Yes	20	34	54	*p* < 0.001
No	55	9	64	
Total	75	43	118	
**Negative Stories**	DA low	DA high	Total	*p* < 0.01
Yes	54	40	94	
No	21	3	24	
Total	75	43	118	
**Negative Experiences**	**Neg. Stories**			
	Yes	No	Total	
Yes	48	6	54	*p* < 0.05
No	46	18	64	
Total	94	24	118	

^1^ Differences assessed using Pearson’s Chi-square test.

**Table 3 dentistry-13-00162-t003:** Mean scores and standard deviations of personality traits for dental anxiety (DA), negative stories, and negative experiences; effect sizes are reported as rank-biserial correlations (r) and significant differences in ratings of personality traits between variable alternatives are marked with asterisks (one-sided Mann–Whitney U-tests).

Dentist Personality Trait	DA ^a^	M (SD)	r	Neg. Exp. ^b^	M (SD)	r	Neg. Stories ^c^	M (SD)	r
Extraversion	low	5.01 (1.05) ***	0.36	yes	4.43 (1.23)	−0.25	yes	4.67 (1.20)	−0.13
	high	4.22 (1.32)		no	4.97 (1.15) **		no	4.92 (1.29)	
Agreeableness	low	5.31 (0.90) ***	0.39	yes	4.69 (1.22)	−0.31	yes	4.95 (1.14)	−0.22
	high	4.54 (1.23)		no	5.31 (0.90) **		no	5.35 (0.88) *	
Conscientiousness	low	5.74 (0.92) **	0.27	yes	5.19 (1.05)	−0.38	yes	5.50 (1.06)	−0.13
	high	5.23 (1.09)		no	5.87 (0.87) ***		no	5.79 (0.77)	
Emotional Stability	low	5.75 (0.93) *	0.19	yes	5.49 (1.00)	−0.22	yes	5.61 (0.95)	−0.19
	high	5.52 (0.93)		no	5.81 (0.85) *		no	5.90 (0.82)	
Openness to Experience	low	4.27 (1.02) *	0.24	yes	3.85 (1.16)	−0.25	yes	4.11 (1.12)	0.00
	high	3.77 (1.13)		no	4.28 (0.98) *		no	3.98 (0.98)	

^a^ low: 75, high: 43; ^b^ yes: n = 54, no: n = 64; ^c^ yes: n = 94, no: n = 24; * *p* < 0.05, ** *p* < 0.01, *** *p* < 0.001.

**Table 4 dentistry-13-00162-t004:** Logistic regression analysis regarding predicting high dental anxiety.

Coefficients	*B*	SE	Odds Ratio (OR)	95% CI OR		*p*
				Lower Bound	Upper Bound	
Constant	−2.09	0.80	0.12			0.01
*Negative Experiences*	2.34	0.49	10.35	3.93	27.25	*<0.001*
Negative Stories	1.26	0.75	3.51	0.81	15.20	0.09
Gender	−1.02	0.55	0.36	0.12	1.06	0.07
*Dentist Extraversion*	−1.08	0.49	0.34	0.13	0.89	*0.03*
*χ* ^2^			43.28			
*df*			4			

Note: Italics denote significant coefficients.

## Data Availability

The raw data supporting the conclusions of this article will be made available by the authors on request.

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
