# Peer review of "The Relationship Between the Perceived Personality Traits of Dentists, Dental Anxiety, Negative Stories, and Negative Experiences with Dental Treatment: A Cross-Sectional Study"

_dentistry, 2025, doi:10.3390/dj13040162_

Round 1
Reviewer 1 Report
Comments and Suggestions for Authors
- I kindly suggest to check the word of "association" instead of "impact" (Line 17,128,169,233).
- Please mention your exclusion criteria( sample selection)
- Paragraphs to consider paraphrasing and cites(70-77)
Author Response
We thank the Reviewer for a thorough and constructive critique of our manuscript. All the changes to the manuscript have been made using “tracked changes” according to the feedback from all reviewers. Below is a point-by-point response to comments from Reviewer 1:
Comment 1. I kindly suggest to check the word of "association" instead of "impact" (Line 17,128,169,233).
Response 1: We have checked to usage of “impact” throughout the manuscript and rephrased where it appeared appropriate in order to match the cross-sectional nature of this study better.
Comment 2. Please mention your exclusion criteria( sample selection)
Response 2: As the current study used a convenience sample of students and the responses were anonymous no explicit exclusion criteria were implemented. Naturally, anyone not enrolled in a student program at the university would not be eligible for participation, but no other efforts were undertaken to exclude participants.
Comment 3. Paragraphs to consider paraphrasing and cites(70-77)
Response 3: We are not sure how to interpret this comment, as the paragraph in question (line numbers 70-77) contains the research hypotheses. We feel that the hypotheses build upon the arguments and former citations without having to repeat these here.
Reviewer 2 Report
Comments and Suggestions for Authors
Dear Authors: I want to congratulate you for the effort employed during the development of the present manuscript.
In my opinion, these are the most noticeable strengths of your study:
- You employ validated measurement tools (IDAF-4C and a modified TIPI) to assess dental anxiety and personality traits.
- The statistical analysis is appropriate, using non-parametric tests for the non-normally distributed data and logistic regression to explore the relative impact of variables on dental anxiety.
- The Abstract, Introduction and Discussion are well-supported by relevant literature on dispositional attribution, dental anxiety, and the significance of negative experiences.
- The findings have practical implications for dental practitioners regarding interpersonal skills and patient experience management.
On the other hand, the main weaknesses are:
- The convenience sample of participants limits generalizability. You acknowledge this limitation, but could further discuss how this sample might differ from the general population regarding dental anxiety and experiences.
- The methodological design cannot establish causality between variables. While the authors discuss a "negative feedback loop", a longitudinal design would be necessary to demonstrate such relationships over time. Please, discuss this concern.
- The key variables are measured with simple yes/no questions without exploring the nature, severity, or recency of these experiences/stories, limiting nuanced analysis. Mention this issue in the Limitations section.
- While the authors explain their adaptation of the TIPI for rating others rather than self-rating, more discussion of the validity of this approach is needed, as this instrument was designed for self-assessment.
- Although the sample is primarily young adults (78% aged 18-25), there is minimal analysis of how age or other demographic factors might influence the findings.
Recommendations:
- Please, describe, in a detailed way, the process of sample size calculation.
- The ethical concerns only appear in the final of the manuscript. Please, include a specific paragraph in the M and M section.
- I recommend providing a study hypothesis after the investigation aims, in the Introduction section.
- Provide evidence for the validity process of the modified TIPI in measuring perceived personality traits of others, or acknowledge this as a methodological limitation.
- Acknowledge more explicitly the limitations of cross-sectional design in establishing the proposed "negative feedback loop."
- Expand the discussion of practical implications for dental practitioners, including specific strategies to address the findings regarding extraversion and its relationship to dental anxiety.
- The statement about "easy to imagine that stereotypically 'extraverted' behaviors are more conducive to communication during dental treatment" (lines 215-218) would benefit from supporting evidence or citations.
Author Response
We thank the Reviewer for a thorough and constructive critique of our manuscript. All the changes to the manuscript have been made using “tracked changes” according to the feedback from all reviewers. Below is a point-by-point response to comments from Reviewer 2:
Recommendations:
Comment 1. Please, describe, in a detailed way, the process of sample size calculation.
Response 1: We have added details about the sample size calculation in the Methods section. Please note that due to the novel nature of the study, the calculation has its limitations.
Comment 2. The ethical concerns only appear in the final of the manuscript. Please, include a specific paragraph in the M and M section.
Response 2: We have added the ethical approvals to the Methods and Materials section of the manuscript.
Comment 3. I recommend providing a study hypothesis after the investigation aims, in the Introduction section.
Response 3: We have provided an overall study hypothesis before the specific research hypotheses (A-C). See lines 74-76.
Comment 4. Provide evidence for the validity process of the modified TIPI in measuring perceived personality traits of others, or acknowledge this as a methodological limitation.
Response 4: We acknowledge this is as a methodological limitation and have added more about this in the paragraph concerning study limitations (see lines 316-323).
Comment 5. Acknowledge more explicitly the limitations of cross-sectional design in establishing the proposed "negative feedback loop."
Response 5: While we propose a negative feedback loop in the introduction as a way of thinking about the relationship between the variables, we have not set out to test this as it would require other study designs. We feel this is quite evident from the research hypotheses, however we have added a cautionary note about causal and/or directional interpretations in the section discussion limitations; see lines 315-316.
Comment 6. Expand the discussion of practical implications for dental practitioners, including specific strategies to address the findings regarding extraversion and its relationship to dental anxiety.
Response 6: We have tried to expand on the practical implications (see lines 301-309). Some of your concerns are perhaps also addressed by the notion of social competence as a key concept to explain positive effects of extraversion, which is presented earlier in the Discussion; see lines 268-271.
Comment 7. The statement about "easy to imagine that stereotypically 'extraverted' behaviors are more conducive to communication during dental treatment" (lines 215-218) would benefit from supporting evidence or citations.
Response 7: We have reformulated the sentence and section mentioned here in order to improve readability and introduce more supporting evidence; see lines 246-256.
Reviewer 3 Report
Comments and Suggestions for Authors
This manuscript discusses a relevant research question to public health and dental public health, considering the increasing focus on patient centered care, trust in healthcare professionals, and barriers to dental care utilization. However, the study has several limitations and methodological concerns that should be addressed before acceptance for publication.
The abstract is generally clear but it lacks numerical data for the key findings e.g., p-values or odds ratios are clearly missing.
University students were the target population and they may have different dental experiences, education levels, and socio-economic backgrounds than the general public which is a clear bias so authors should justify why this sample was chosen
The sample size of 118 participants is relatively small, especially considering subgroup comparisons and power analysis was NOT reported to justify the adequacy of the sample size.
There is no mention of pilot testing of the translated version of the study tool in Norwegian.
What is the rationale for the dichotomization of dental anxiety scores (low and high). No mention of cut-off points in the manuscript, isn’t it better to split it in low medium and high?
Were correlations between independent variables checked before including them in regression models? Also, effect sizes e.g., Cohen’s d, odds ratios with confidence intervals are not calculated/reported.
The discussion should argue on why negative stories had limited influence.
In addition, expand your discussion to compare results with previous studies on medical professionals as well owing to similarity in their professional duties.
The discussion section should be proofread as it contains some long and complex sentences.
Comments on the Quality of English Language
English language expression is overall fine but the discussion section should be proofread as it contains some long and complex sentences.
Author Response
We thank the Reviewer for a thorough and constructive critique of our manuscript. All the changes to the manuscript have been made using “tracked changes” according to the feedback from all reviewers. Below is a point-by-point response to comments from Reviewer 3:
Comment 1. The abstract is generally clear but it lacks numerical data for the key findings e.g., p-values or odds ratios are clearly missing.
Response 1: We acknowledge that the original abstract was lacking in numerical information as we made an effort to summarize key findings in plain language, so that interested readers could judge whether the paper is of interest for further reading. While we find it challenging to add all numerical indices without expanding the abstract beyond the confines set by the publisher we have now added the odds ratios and p-values connected to the regression analysis.
Comment 2. University students were the target population and they may have different dental experiences, education levels, and socio-economic backgrounds than the general public which is a clear bias so authors should justify why this sample was chosen
Response 2: The sample is clearly stated to be a convenience sample which is chosen due to convenience and availability. We think this has been clearly stated in the text, but we have attempted to be clearer about this in the limitations; see paragraph (see lines 310-316).
Comment 3. The sample size of 118 participants is relatively small, especially considering subgroup comparisons and power analysis was NOT reported to justify the adequacy of the sample size.
Response 3: We have added a clarification about power analysis in the Materials and Mehtods section (see lines 132-136). Note that since the modified TIPI represents a novel approach and there are no studies to our knowledge using the TIPI in relation to the topics addressed in the current manuscript, the calculations were aimed at identifying relationships between negative stories, negative experiences and dental anxiety, and did not directly relate to the analyses related to personality attributions, which is a limitation. We have tried to be specific about this limitation in the amended text.
Comment 4. There is no mention of pilot testing of the translated version of the study tool in Norwegian.
Response 4: We interpret this comment to address the use of the IDAF-4C. The version we have used is the same as has been used previously by Norwegian researchers in this field, see e.g. reference 18 which describes the translation process under “outcome measures”. No additional pilot testing was performed with regards to the current study. To distinguish more clearly between the original reference and the study using a translated version we have now separated the references (references 18 and 19; see lines 98-99; note that this is not marked as a change due to the reference software used not being compatible with tracked changes. For sake of completeness, we have also added a reference to the Norwegian validation of the unmodified TIPI (ref 22), although we are not directly utilizing this measure.
Comment 5. What is the rationale for the dichotomization of dental anxiety scores (low and high). No mention of cut-off points in the manuscript, isn’t it better to split it in low medium and high?
Response 5: We have described using the mean scores as a dichotomization point for the IDAF-4C in the Methods section. With regards to cut-offs these are typically used to determine clinically relevant dental anxiety vs non-clinically relevant dental anxiety, which was not the intention here. Rather, we aimed at testing for associations between anxiety scores and the independent variables, and for this reason we chose to use a dichotomization approach rather than the cut-offs described in several other studies. Using cut-offs for dental phobia for instance would have resulted in a smaller percentage of the sample in this group (according to some studies around 10%), which would have made the regression analysis difficult to interpret. With regards to using more detailed categories (low, medium, high) we chose to simplify the analysis as much as possible, also given that the sample size (as mentioned above) is quite low and the meaningfulness of three categories vs two are not clear.
Comment 6. Were correlations between independent variables checked before including them in regression models? Also, effect sizes e.g., Cohen’s d, odds ratios with confidence intervals are not calculated/reported.
Response 6: We did not perform specific correlational analyses to check correlation between independent variables prior to the regression analysis, however the initial analyses (Chi-square) strongly indicate associations between these. We did however look at multicollinearity parameters (VIF) as part of the analyses, which do not indicate issues with multicollinearity. Also, we have added effect sizes to the results. Here, we have used the rank-biserial correlation, which is to our knowledge the most common way to report effect size for Mann-Whitney tests. Also, confidence intervals have now been added to the regression model; see Table 4. Note that we found an unfortunate error in Table 4 where the Wald-index had been reported instead of the OR for one variable; this has been corrected also.
Comment 7. The discussion should argue on why negative stories had limited influence.
Response 7: We have expanded the discussion on this topic; see lines 280-287.
Comment 8. In addition, expand your discussion to compare results with previous studies on medical professionals as well owing to similarity in their professional duties.
Response 8: We have attempted to include more relevant research about other healthcare professions in the discussion. However, we find that most other studies have used methodological approaches that limit their usability here.
Comment 9. The discussion section should be proofread as it contains some long and complex sentences.
Response 9: We agree, and we have attempted to edit the discussion with regards to readability.
Comment 10. Comments on the Quality of English Language: English language expression is overall fine but the discussion section should be proofread as it contains some long and complex sentences.
Response 10: See Response to Comment 9.
Reviewer 4 Report
Comments and Suggestions for Authors
The paper entitled “The relationship between perceived personality traits of dentists, dental anxiety, negative stories, and negative experiences with dental treatment: A cross-sectional study” is an interesting contribute that aims explores the impact of dental anxiety, negative dental stories and negative dental experiences on the perception of dentists’ personality traits. The work provides interesting data and could be considered of interest for the readers. However before it could be considered valid for publication requires some corrections.
INTRODUCTION
Overall well structured, it provides all the information necessary to understand the scientific background, the knowledge gap and the objectives of the study.
MATERIAL AND METHODS
The scientific methodology used were described in a clear and exhaustive manner. There iconography is quite poor and should be improved. Il could be of interest an image of the questionnaire used for the investigation in order to know the 13 questions submitted to the students.
RESULTS
The results are described in a precise and detailed manner; graphical representation is well executed and allows a faster understanding of the results achieved in the study. Statistical analysis performed is correct.
DISCUSSION
The discussion of the results is on the whole well articulated
CONCLUSION
Conclusions section is not present. In my opinion this section should be added and repor,t preferably with a bulleted list, the key results of the study.
Author Response
We thank the Reviewer for a thorough and constructive critique of our manuscript. All the changes to the manuscript have been made using “tracked changes” according to the feedback from all reviewers. Below is a point-by-point response to comments from Reviewer 4:
Comment 1. INTRODUCTION: Overall well structured, it provides all the information necessary to understand the scientific background, the knowledge gap and the objectives of the study.
Response 1. Thank you for the positive feedback.
Comment 2. MATERIAL AND METHODS: The scientific methodology used were described in a clear and exhaustive manner. There iconography is quite poor and should be improved. Il could be of interest an image of the questionnaire used for the investigation in order to know the 13 questions submitted to the students.
Response 2. We are not sure what the iconography refers to specifically as this section uses few symbols or similar (compared to e.g. the Results section); please advice us further here. Concerning the questionnaire, we are obviously happy to include this in some form, for instance as supplementary materials. We feel that pasting it as an image or figure into the text would be counterproductive.
Comment 3. RESULTS: The results are described in a precise and detailed manner; graphical representation is well executed and allows a faster understanding of the results achieved in the study. Statistical analysis performed is correct.
Response 3. Thank you for the positive feedback.
Comment 4. DISCUSSION: The discussion of the results is on the whole well articulated
Response 4. Thank you for the positive feedback.
Comment 5. CONCLUSION: Conclusions section is not present. In my opinion this section should be added and repor,t preferably with a bulleted list, the key results of the study.
Response 5: We have used the template provided by the publisher, which do not include a conclusion section. The publisher (MDPI) states that it is recommended only if the discussion section is “unusually long or complex” and we do not believe that we fall into either category.
Round 2
Reviewer 3 Report
Comments and Suggestions for Authors
The responses from the reviewers are comprehensive and the changes made in the revised manuscript are quite satisfactory. This looks in good shape to me now.